# Laser Photobiomodulation (PBM)—A Possible New Frontier for the Treatment of Oral Cancer: A Review of In Vitro and In Vivo Studies

**DOI:** 10.3390/healthcare9020134

**Published:** 2021-01-29

**Authors:** Alessandro Del Vecchio, Gianluca Tenore, Maria Clotilde Luzi, Gaspare Palaia, Ahmed Mohsen, Daniele Pergolini, Umberto Romeo

**Affiliations:** Department of Oral Sciences and Maxillofacial Surgery, Sapienza University of Rome, 00161 Rome, Italy; alessandro.delvecchio@uniroma1.it (A.D.V.); gianluca.tenore@uniroma1.it (G.T.); luzi.1758422@studenti.uniroma1.it (M.C.L.); ahmed.mohsen@uniroma1.it (A.M.); daniele.pergolini@uniroma1.it (D.P.); umberto.romeo@uniroma1.it (U.R.)

**Keywords:** chemotherapy, contraindications, dysplasia, head–neck carcinoma, low-level laser therapy, neoplasia, oral cancer, potentially malignant disorders, photobiomodulation, radiotherapy

## Abstract

The treatment of oral squamous cell carcinoma (OSCC) is particularly complex due to its aggressive behavior, location, the patient’s age, and its spread at diagnosis. In recent years, photobiomodulation (PBM) has been introduced in different medical fields; however, its application, in patients suffering from OSCC for palliative support or to induce analgesia, has been hotly debated due to the possibility that the cell growth stimuli induced by PBM could lead to a worsening of the lesions. The aim of this study is to review the literature to observe the available data investigating the effect of PBM on cancer cells in vitro and in vivo. A review was conducted on the PubMed and Scopus databases. A total of twelve studies met the inclusion criteria and were therefore included for quality assessment and data extraction. The analysis showed that the clinical use of PBM is still only partially understood and is, therefore, controversial. Some authors stated that it could be contraindicated for clinical use in patients suffering from SCC, while others noted that it could have beneficial effects. According to the data that emerged from this review, it is possible to hypothesize that there are possibilities for PBM to play a beneficial role in treating cancer patients, but further evidence about its clinical efficacy and the identification of protocols and correct dosages is still needed.

## 1. Introduction

Oral squamous cell carcinoma (OSCC) is a public health problem and a major psychological threat and adaptation issue for patients. It is a disease in which dentists and dental hygienists can play a decisive role in early detection and supportive disease management. In recent years, there has been an increase in the incidence of OSCC due to an increase in tobacco and alcohol consumption, especially in females and in the younger population. The 90% of the lip and oral cancer is represented by OSCC. In addition, OSCC is the twelfth most prevalent cancer in all the world. The estimated number of new cases with oral and lip cancer was 354,864 worldwide and 3967 in Italy. Furthermore, the estimated number of deaths in 2018 due to oral and lip cancer were 177,384 all over the world and 1489 in Italy [1,2,3]. The prognosis for OSCC is greatly influenced by the stage at diagnosis, which is still often too late for effective treatment. Even though the surgical techniques and the diagnostic tools have been enhanced over the years, there has been no beneficial effect in terms of OSCC prognosis, so that the differences in mortality rates in the different geographical areas are attributable to variations in the exposure to risk factors [2]. OSCC treatment is mainly based on demolitive surgery and chemo- and radiotherapy in different combinations—treatments that produce, at the same time, severe side effects that have a negative impact on the quality of the remaining life of the patients. It has become largely evident that adjunct therapies to manage the consequences of the therapeutic protocols for these patients are no longer deferable, and thus, many authors have suggested the possibility of investigating the potential beneficial effects of photobiomodulation (PBM) in OSCC patients [4].

PBM is the application of red or near-infrared (NIR) light to heal, restore, and stimulate physiological processes, and to repair damage caused by injuries or diseases [5]. The “optical window” in which the effective penetration of light into tissues is maximized is between approximately 600 nm and 1200 nm. Low-energy laser light produces photochemical effects whereby it penetrates the mucosa without overheating or producing other side effects. The emitted photons reach the inner mitochondrial membrane and the light acts on cytochrome c-oxidase (COX) with the consequent production of adenosine triphosphate (ATP), reactive oxygen species (ROS), and the release of nitric oxide (NO) [6]. Due to this mechanism, PBM improves the vital capacity of cells, induces growth factor production, and enhances the motility and viability of the irradiated cells.

PBM follows the rules of the ‘‘biphasic dose–response’’ curve or the Arndt–Schulz curve, just like conventional drugs do [7]. This principle states that there are optimal parameters (energy density or power density) that provide a benefit to the irradiated tissues, but if these parameters are significantly exceeded, the irradiation could lead to harmful effects. This phenomenon is also called ‘‘hormesis’’ and has been widely reviewed by Calabrese and Mattson [8] and by Calabrese and Baldwin [9].

Even though studies on PBM began in the early 1960s and a large number of them describe its beneficial effects in the treatment of various medical conditions, especially in dermatology [10], many aspects of the clinical use of PBM are still only partially understood, and, to date, the same limits apply in terms of its therapeutic possibilities.

Due to the proliferative effect induced on cells, the applications of PBM in OSCC patients have been greatly debated over the years: Sroka et al. demonstrated that PBM stimulated tumor cell growth in cell culture studies [11]; Sperandio et al. stated that it could increase the aggressiveness of some cancer cells [12]; Navratil et al. affirmed that the clinical application of PBM is totally contraindicated in cancer patients [13]. However, not all experimental studies have led to the same findings; for example, Zecha et al. showed that PBM was highly effective in mitigating the numerous side effects that occur because of cancer therapy [14,15]. In addition, many contributions have supported the hypothesis that, just like in many other diseases, PBM can really have a beneficial effect if used on the neoplasia site, and that there are three possible mechanisms for this finding: (1) a direct effect on neoplastic cells [16]; (2) the selective inhibition of malignant cells [17]; (3) stimulation of the immune system [18].

As the International Agency for Research on Cancer (IARC) stated in 2018, the incidence and mode of OSCC are increasing, and the impact of the phenomenon could be alarming. Traditional therapies such as chemotherapy and radiotherapy are sometimes not sufficient to block the neoplasm, inducing, in the meanwhile, severe side effects such as tissue necrosis or mucositis [19]. The beneficial effect of PBM on irradiated tissues has been largely demonstrated over the years in many pathologic conditions, but its use in the treatment of OSCC or dysplastic lesions has been hampered due to its proliferative effects on the irradiated tissues. However, many authors have stated that it can directly damage the tumor mass, enhance other cancer therapies, stimulate the host immune system, and increase survival rates in OSCC patients [16,17,18].

The aim of this study is to review the literature to observe the available data investigating the effect of PBM on cancer cells and lesions in vitro and in vivo. The different laser protocols were studied in order to evaluate the wavelength (nm), the power (W), the PBM delivery technique, the amount of delivered energy (J), the energy density (J/cm^2^), the delivery times (sec), and the delivery schedules (hours, days, weeks).

## 2. Materials and Methods

This review was developed following the parameters of Preferred Reporting Items for Systematic Reviews and Meta-analyses (PRISMA) guidelines. It was registered at PROSPERO “the international prospective register of systematic reviews” with registration number CRD42021224774. The methods and criteria for inclusion were selected based on the PRISMA guidelines.

### 2.1. Eligibility Criteria

#### 2.1.1. Inclusion Criteria

All studies concerning PBM applied on cancer were included in this review that met the following requirements; being carried out on cancer cells and/or oral dysplastic cells, performed in vivo or in vitro, investigated the PBM effect, and presented the evidence of clinical efficacy. Only articles in the English language were included in this study.

#### 2.1.2. Exclusion Criteria

Studies with a lack of evidence of clinical efficacy, no PBM application, no available experimental data, and items not in English were excluded. Reviews (narrative and systematic), case reports, abstracts, and letters to editors were also excluded from the study.

### 2.2. Search Strategy

An electronic search was conducted using the PubMed and Scopus databases. Appropriate free-text keywords and controlled vocabulary terms were initially extracted from some key articles and were used in a series of pilot electronic searches. The key terms were further enriched as additional terms came up, and the electronic search was repeated each time. The used keywords were combined with the Boolean term “AND” and “OR”: oral cancer, neoplasia, dysplasia, potentially malignant lesions, Photobiomodulation, low-level laser therapy, and contraindications. In addition, the reference lists of the included studies were manually searched to identify other publications that were not retrieved from the primary search of the Scopus database. The search for articles of interest began in September 2018 and completed in December 2020.

### 2.3. Study Selection

In the first stage, the titles and abstracts of the identified articles were independently screened through the application of the abovementioned inclusion and exclusion criteria by two reviewers (M.C.L. and G.P.). In the second stage, the two reviewers independently read the full text of each resulted article to identify those that could be of benefit to the review. Those articles that lacked any of the eligibility criteria were excluded. In case of disagreements between the two reviewers, a discussion with a third reviewer (A.D.V.) was performed.

### 2.4. Data-Collection and Synthesis Process

Two authors (M.C.L. and G.P.) screened the full text independently to extract the data from each eligible study. Disagreements were resolved through arbitration by a third reviewer (A.D.V.). The extracted data of each individual study were the author/year, the study type, the laser type, the sample characteristics, the method of evaluation of PBM effect, the main outcomes, conclusions, and the overall PBM effect. In addition, the PBM parameters were extracted from each individual study; including the wavelength (nm), the type of emitter, power (mW), energy per point (J), irradiation time (sec), spot size (cm^2^), energy density (J/cm^2^), total energy (J), delivery technique, and schedule.

A narrative and tabular synthesis of data were performed for all the included studies. The overall effect of PBM was described in two possible outcomes; (1) positive outcome where the PBM showed having an inhibitory effect on the cell proliferation and viability and/or on the aggressiveness and severity of cancer cells or lesions, (2) negative outcome where the PBM showed having a stimulatory effect on the cells proliferation and/or on the aggressiveness and severity of cancer cells or lesions. The included studies were divided based on these two outcomes and described narratively.

### 2.5. Reporting Quality and Risk of Bias Assessments

Different assessment tools were used to evaluate the reporting quality and risk of bias at the study level of each eligible study. Each selected study was subjected to the appropriate tool of assessment and scored independently by two reviewers (M.C.L. and G.P.). Conflicts were resolved through arbitration by a third reviewer (A.D.V.).

For in vivo studies, the risk of bias tool of the SYstematic review Centre for Laboratory animal Experimentation (SYRCLE) was used. It consists of 10 items to evaluate 6 types of bias. Each item was scored with low, unclear, or high risk [20]. For in vitro studies, based on a systematic review of in vitro studies, the authors developed and used predefined criteria due to the absence of a standard quality tool and risk of bias tool. The types of the assessed biases with these developed criteria were selection, performance, and detection bias [21]. The scores for this assessment were classified into high and low risk. In case of a lack of details to assess the bias, the score “Risk unknown” was used.

In addition, the reporting quality was assessed and determined for both kinds of studies (in vitro and in vivo) through predefined criteria to assess the lack of reproducibility, which were also based on the same systematic review [21]. The scores for this assessment were “Reported”, “Not clearly reported”, and “Not reported”. Table 1 shows the developed reporting quality scheme (for both the in vivo and in vitro studies) and the developed risk of bias scheme (only for the in vitro studies).

The heterogeneity among studies was observed; therefore, the meta-analysis could not be performed.

## 3. Results

A total of 1137 studies were identified from the PubMed database search between 1992 and 2020. In addition, a total of 78 studies were identified from the Scopus database search between 2005 and 2020. After screening the titles, abstracts, and reference lists, a total of 157 articles were subjected to full-text screening. A total of 12 articles met the inclusion criteria and were included for data extraction and synthesis. The reasons for the exclusion of 145 studies were that not using PBM, using other kinds of laser application such as Photodynamic therapy (PDT), no complete experimental available data, no core parameters for evaluation, presence of items not in English, and/or lack of evidence of clinical efficacy. Figure 1 shows the flowchart of the evaluation process of the publications.

The distribution of the included studies in this review was as follows; seven in vitro studies, three in vivo studies, and two both in vitro and in vivo. All the included studies were subjected to qualitative analysis through the assessment of reporting quality and risk of bias assessments. Figure 2, Figure 3 and Figure 4 show the resulted qualitative analysis of all the included studies.

There was diversity among the included studies in the PBM parameters, sample type and size, and the methodological approach. This diversity hindered the authors from performing the meta-analysis. It was decided to perform a tabular and narrative review based on this systematic approach.

In the nine in vitro studies, several types of cell lines were employed; including human SCC of gingival mucosa (ZMK1), KB cells, melanoma cells (B16F10), human SCC (SCC25 and SCC9), human SCC cell line (CAL27), and SCC cells originated from the tongue (TSCC-1). Several methods of evaluation were employed to observe the effect of PBM; including mitosis rate, cell cycle distribution, apoptosis assay, proteins analysis, cell invasion analysis, and ATP production assay.

In the in vivo studies, different animal tumor models were employed. Oral carcinogenic-induced tumor models were employed in two of them using 4-nitroquinoline-1-oxide (4-NQO) in mice or 7, 12-dimethylbenz[a] anthracene (DMBA) in hamsters. In one study, the nonmelanoma ultraviolet (UV)-induced skin cancer model was used in SKH mice. In the other two studies, the animal tumor models were achieved through the injection of melanoma cells (B16F10) in mice or anaplastic thyroid cancer cell line FRO in the thyroid gland of mice. All the details of the included studies in this review including the main outcomes and conclusions were summarized in Table 2.

Regarding the overall effect of PBM, the inhibitory effect of PBM (positive outcome) was reported in six studies [11,18,25,26,29,30]. The other six studies reported a stimulatory effect (negative outcome) [12,22,23,24,27,28]. They were distributed as follows; two studies reported an increase in the invasion and aggressiveness with PBM [12,24], two studies reported the increase in cell proliferation and tumor growth [22,23], and the other two studies reported both stimulatory effects of PBM (proliferation and aggressiveness) [27,28]. A detailed description of the included studies grouped based on the outcomes (positive and negative) are presented in the following sections.

### 3.1. Studies that Demonstrate Inhibitory Effect with PBM (Positive Outcome)

There were two in vitro studies that studied the effect of PBM on the proliferation of cells and demonstrated positive outcomes (inhibitory effect) [11,26]. Sroka et al. studied the effect of different wavelengths with different energy densities on the mitosis rate of different human cell lines including normal cells and the human SCC of gingival mucosa (ZMK1). A slight decrease in the mitosis rate of ZMK1 was observed with the increase in the irradiation energy independent of the wavelength. Comparing to controls without dependence on wavelength, a slight decrease in mitosis rate at irradiation of 20 J/cm^2^ was observed [11].

Schartinger et al. used a 660 nm laser to investigate the effect of PBM on human SCC cells (SCC25) and comparing the results with human normal cells. These were through cell proliferation assay by MTT, cell cycle analysis by propidium iodide and FACS analyses, and apoptosis assay by annexin V-fluorescein isothiocyanate (V–FITC) flow cytometry analysis. The results revealed that PBM showed a significant decrease in cell proliferation and percentage of G1-phase cells, and a significant increase in the percentage of S-phase cells when compared to controls. In addition, a proapoptotic effect was observed with PBM on SCC25 [26].

In addition, two in vitro studies investigated the effect of PBM on the progression and severity of malignant cells and demonstrated positive outcomes (inhibitory effect) [29,30]. Takemoto et al. studied the effect of LED-based PBM with different energy densities (3, 6, 9, 12, 24, 36 J/cm^2^) on the progression of malignant invasion of human SCC cell lines (CAL27) seeded over normal stromal of gingival fibroblasts. Additionally, they analyzed the cell counts by flow cytometry, viability by the MTT assay, and apoptosis by annexin V–FITC flow cytometry analysis of the culture model at an energy density of 36 J/cm^2^. It was found that after 72 h of treatment, PBM inhibited the expansion of colonies of cells. At high doses of PBM (36 J/cm^2^), a general advantage on the stromal fibroblasts over cancer cells was observed with regard to the cell viability, apoptosis, and death assays [29].

Shirazian et al. studied the effect of PBM on the proliferation and invasion of SCC cells originated from the tongue (TSCC-1). Two different wavelengths (660 and 810 nm) were used with two different parameters. Cell proliferation was analyzed by MTT assay, while the invasion analysis was carried out by evaluating six markers. Flow cytometry was employed to assess cyclin D1, β- catenin, E-cadherin, and Matrix Metallopeptidase-9 (MMP-9) markers level. Real-time polymerase chain reaction (RT-PCR) was employed to assess Ki67 and vascular endothelial growth factor (VEGF) expression levels. In 810 nm groups (100 and 200 mW), higher percentages of cyclin D1 and MMP-9 were observed. A significant decrease in VEGF marker in the 810 nm group at the power of 200 mW. In 660 nm groups (40 and 80 mW), higher percentages of β-catenin and E-cadherin were observed. No differences were observed among groups for the Ki67 marker. They concluded that PBM (660 nm with 80 mW and 810 nm with 200 mW) at 4 J/cm^2^ can have an inhibitory effect on the proliferation of OSCC. They also recommended considering time as an important factor in the effectiveness of PBM [30].

Two in vivo studies demonstrated positive outcomes [18,25]. One of them showed a neutral effect of PBM without inhibition or stimulation of malignant lesions and is safe for application [25]. Whereas, Myakishev-Rempel et al. treated SKH mouse nonmelanoma UV-induced skin cancer models with LED-based PBM twice daily for 37 days (at 2.5 J/cm^2^), and photographic measurements did not reveal a measurable effect of PBM on tumor growth [25]. The other in vivo study showed an inhibitory effect of PBM on tumor progression [18]. The investigators studied the effect of three different protocols of PBM on several cultured cells and on oral carcinogenic-induced tumor models in mice using 4-NQO. Macroscopic, histological, and flow cytometry analysis were performed. The study revealed a reduction of the tumor progression associated with the secretion of type I interferons from T lymphocytes and dendritic cells. In addition, a decrease in the angiogenic macrophages was observed in the tumor mass with a promotion of the vessel normalization [18].

### 3.2. Studies that Demonstrate a Stimulatory Effect of PBM (Negative Outcome)

There were two studies that demonstrated the negative outcome and stimulating effect on the proliferation of malignant cells and increase in tumor volume [22,23]. In one of them, the effect of PBM was studied on both cultured cells (melanoma cells (B16F10)) and melanoma cells in an animal model (in vivo) [23]. A diode laser (660 nm) was employed with two different energy densities (150 and 1050 J/cm^2^). In the in vitro section, cell viability and cell cycle changes by Tripan Blue, MTT, and cell quest histogram were performed for assessing the effect of PBM. While, in the in vivo, the tumor volume and histological characteristics were the evaluation methods. The authors observed an almost absence of statistical difference between in vitro groups. However, they observed a stimulatory effect of PBM (negative outcome) in the in vivo group of high doses (1050 J/cm^2^) with a significant increase in the tumor volume, blood vessels, and cell abnormalities [23]. The other study was an in vitro study that investigated the effect of two different diode lasers (685 and 830 nm) on the cellular viability (assessed by MTT assay) of KB cells. They found that the test group irradiated with 830 nm showed a significant increase in proliferation when compared to the other test group (685 nm). PBM in both test groups significantly influenced the cellular viability when compared to controls. They concluded that PBM has a bio-modulatory effect on KB cells [22].

In total, four studies showed the negative outcome and stimulatory effect of PBM on the invasiveness and aggressiveness of malignant cells and lesions [12,24,27,28]. Two of them were conducted on cultured cells (in vitro) and studied the effect of PBM through the analysis of protein markers [12,27]. Sperandio et al. studied the possible effect of PBM using 660 nm and 780 nm (at 40 mW, and 2.05, 3.07, or 6.15 J/cm^2^ for each) in increasing the aggressiveness of oral dysplastic cells (DOK) and oral cancer cells (SCC9 and SCC25). Protein analysis was performed by Western blot and immunofluorescence. The study revealed that PBM was able to significantly modify the expression of progression- and invasion-related proteins in all cell lines and also aggravate the cellular behavior of SCC25. Whereas, it was observed with both wavelengths an increased expression of cytosolic (p-Akt) proteins, ribosomal protein (pS6), and cell-cycle progression regulator (cyclin D1) producing an aggressive isoform of Heat-shock protein 90 (Hsp90) [12]. Gomes Henriques et al. achieved the same negative outcome of PBM. They also studied the effect of PBM on the protein expression analysis of human SCC of the tongue (SCC25) using a 660 nm laser with two energy densities (0.5 and 1.0 J/cm^2^). They observed almost similar results, where PBM with the energy density of 1.0 J/cm^2^ showed a significant increase in the expression of cyclin D1 and nuclear β-catenin, and promotion of invasion through the reduction of E-cadherin and induction of MMP-9 expression [27].

Two in vivo studies demonstrated the negative outcome of PBM through the increase in aggressiveness and severity of malignant lesions [24,28]. de C Monteiro et al. assessed the effect of PBM (660 nm) through the histological analysis on oral carcinogenic-induced tumor models in hamsters using DMBA. The test group (with PBM) showed a significant difference in the amount of poorly differentiated tumors when compared to other groups without PBM, where the histological analysis of the test group was 40% well-differentiated SCCs, 40% poorly differentiated SCCs, and 20% moderately differentiated SCCs [24].

Rhee et al. investigated how a single dose of PBM using a 650 nm (at 15 and 30 J/cm^2^) could cause an increase in the aggressiveness of anaplastic thyroid cancer [28]. The evaluation was through the assessment of tumor volume, histological evaluation, and assessing the overproliferated FRO cells using immunohistochemical staining with hypoxia inducible factor 1α (HIF-1α), p-Akt, VEGF, and transforming growth factor β1 (TGF-β1). It was found that PBM caused an elevation of HIF-1α and p-Akt, and a decrease in TGF-β1 expression that led to the loss in the cell cycle regulation. It was concluded that these effects may cause an over-proliferation and angiogenesis of cancer cells and PBM may cause aggressiveness of cancer through TGF-β1 and Akt/HIF-1α cascades [28].

### 3.3. Laser Parameters of the Included Studies

With regard to the PBM parameters, there were a diversity of the PBM parameters. The type of emitter was reported in almost all the studies. Diode lasers were the commonly investigated lasers for PBM; including Gallium Aluminium Arsenide (GaAlAs), Indium–Gallium–Alluminium-Arsenide Phosphide (InGaAlAsP), Indium–Gallium–Aluminum Phosphide (InGaAlP), and Gallium Arsenide (GaAs) lasers. Light-Emitting Diodes (LED) were utilized in two studies [25,29]. Neodymium-doped Yttrium Aluminum Garnet (Nd:YAG), Kr+, and Ar+- pumped tunable dye lasers were tested in one study [11]. The investigated wavelengths ranged from 410 to 1064 nm. A total of 11 studies reported the energy density with a range of 0.5 and 1050 J/cm^2^. The investigated power ranged from 30 mW to 2.5 W. Some important missing data of PBM parameters were observed. All the PBM parameters of all the included studies were summarized in Table 3.

## 4. Discussion

The purpose of this review was to examine the results of studies on PBM applied in the neoplastic site. While some authors have stated that it may be contraindicated for clinical use in patients with neoplasms, others find that it may have a beneficial effect on carcinomas. There were a diversity of the PBM parameters. A wide range of wavelengths was employed in the included studies. Almost all of them were in the spectral range of the “optical window” (650–950 nm). This was expected because lasers below 650 nm are strongly absorbed mainly by hemoglobin and over 950 is strongly absorbed by water, which correspondingly may cause overheating of tissues. With the spectral range of the optical window at low energy, the penetration of light is maximized through the mucosa without overheating, reaching the inner mitochondrial membrane, and resulting in photochemical effects [10,31].

From this experience, it emerged that there are three possible ways in which PBM could have a beneficial effect on cancer. The first is the direct effect on neoplastic cells that can occur by exploiting the biphasic dose–response curve to “overdose” cancer cells. Several applications of low- and high-fluency PBM on cancer stem cells (CSCs; i.e., on malignant cells believed to come from either genetically or epigenetically altered healthy stem cells) have been performed to evaluate the different cellular responses [32,33,34,35,36]. Since normal cells and CSCs share similarities in their mitochondrial content, chromophores located in the inner membrane of CSCs are expected to play a photoacceptor role like that found in normal cells [16].

The light photon coming from the laser is absorbed by the COX of the respiratory chain. The energy of the photons donated by visible red light at low fluence is sufficient to dissociate NO from COX and improve the COX reduction capacity, which eventually leads to CSC proliferation through ATP, cAMP, and moderate ROS production. Conversely, the energy of the photons donated by the visible high-fluence red light is sufficient to reduce the COX reduction capacity that leads to the massive conversion of dioxygen into ROS that causes programmed cell death. When using low fluences of between 5 and 10 J/cm^2^ and 20 J/cm^2^ with wavelength between 600 and 800 nm, a significant increase in both viability and proliferation in CSCs was observed. However, a statistically significant reduction in viability and proliferation that goes hand in hand with an increase in apoptosis could be observed following exposure to 40 J/cm^2^. Thus, it is evident that the biostimulating or bioinhibiting effect of PBM is based on the fluence and wavelength of light. In fact, this is in accordance with Arndt–Schultz’s Law, since weak stimuli (referring to the time of irradiation or to the light dose) increase physiological activity, medium stimuli inhibit activity, and very strong stimuli block activity [16,32,33,34,35,36,37,38,39,40,41,42,43,44,45].

Wu et al. [32] experimented with a new mode of cancer treatment using high fluence low-power laser irradiation (HF-LPLI) in experimental guinea pigs. It has been demonstrated that HF-LPLI (633 nm, 120 J/cm^2^) could induce cancer cell apoptosis through an intrinsic mitochondrial/caspase-3 pathway by triggering the generation of ROS. It has also been found that HF-LPLI induces ROS-mediated mitochondrial permeability transition (MPT) through the mitochondrial pathway. Another pro-apoptotic signaling pathway, including inactivation of the protein kinase B-glycogen synthase 3 beta on HF-LPLI, has also been explored [33].

In addition, in another study by Wu et al. it was found that HF-LPLI in the red light range (635 nm) could selectively photo-inactivate its endogenous COX photoacceptor to generate a mitochondrial superoxide anionic burst (O2−), causing oxidative damage to tumor cells. This mitochondrial phototherapy achieves sufficient antitumor effectiveness without the administration of exogenous chemicals [36]. Although the initial mechanism involved in HF-LPLI-induced ROS generation is still unknown, these reports suggest that higher-dose PBM can be used for tumor therapy through laser and mitochondrial targeting.

The second mechanism is based on the exploitation of a differential effect of PBM on malignant cancer cells and on normal healthy cells. For this to happen, it is necessary to combine PBM with additional cytotoxic and antitumor therapies to increase the killing of tumor cells while protecting normal healthy cells. These considerations are related to the Warburg effect, with which the mitochondria of tumor cells change their metabolism to perform aerobic glycolysis instead of oxidative phosphorylation (OXPHOS). The consequences of the Warburg effect are that malignant cells and normal cells can behave very differently in response to PBM. In tumor cells, where the ATP intake is rather limited, the ATP boost given by PBM can allow tumor cells to respond to pro-apoptotic cytotoxic stimuli with cell death programs (apoptosis), which are highly energy-dependent (i.e., they require a lot of ATP). On the contrary, in normal healthy cells with an adequate ATP intake, the effect of PBM produces an explosion of ROS that could induce protective mechanisms and reduce the harmful effects of cancer therapy on healthy tissues [17,46,47,48,49,50,51,52,53,54,55,56,57,58,59,60,61,62,63].

There are some published studies that suggest that it may indeed be the case in some anticancer strategies, such as reports in which PBM can potentiate the killing of cancer cells with PDT and also with radiotherapy [52]. Tsai et al. evaluated whether PBM could protect cells from cytotoxicity due to PDT, or vice versa, if PBM improved the efficacy of mono-L-aspartyl chlorin(e6) (NPe6)-mediated PDT (NPe6 is a lysosomal localizing photosensitizer (PS)). The idea was that the increase in ATP could lead to better absorption of Npe6 cells by the energy-dependent endocytosis process and also to more efficient apoptosis. To conduct the experiment, the human osteosarcoma cell line (MG-63) was subjected to 1.5 J/cm^2^ of 810 nm NIR followed by the addition of 10 μM Npe6. After 2 h of incubation, 1.5 J/cm^2^ of 652 nm red light was applied per PDT. It was found that PDT combined with PBM led to a higher cell death rate and an increase in intracellular ROS compared to PDT alone. Taken together, these results suggest that PBM potentiates Npe6-mediated PDT through increased ATP synthesis induced by NIR irradiation, and it is a potentially promising strategy that could therefore be applied to potentiate clinical PDT and also to many other cytotoxic carcinoma therapies that require medication and effective apoptosis [52].

Several studies demonstrated a possible another mechanism, where PBM could be beneficial for cancer patients through stimulating the immune system to fight cancer [64,65,66]. Petrellis et al. developed the hypothesis that PBM could induce oxidative stress in tumor tissues that lead to stimulating the generation of pro-inflammatory markers that interfere with tumor progression. They carried out a study to evaluate the levels of pro-inflammatory mediators and gene expression of inflammatory markers after PBM application on 72 Wistar rats to which Walker Tumor 256 (TW-256) cells were inoculated. PBM therapy was carried out with a 660 nm, at the power of 100 mW, with different energies (1 J–35. 7 J/cm^2^, 3 J–107.14 J/cm^2^, and 6 J–214.28 J/cm^2^) three times every 2 days from the 14th day of tumor onset. Although the tumor response was not directly measured, they found that the lowest dose (35.7 J/cm^2^) produced significant increases in IL-1β, COX-2, and inducible nitric oxide synthases (iNOS), and significant reductions in IL-6, IL-10, and tumor necrosis factor-alpha (TNF-α), thus concluding that 35.7 J/cm^2^ was able to produce cytotoxic effects from ROS generation causing acute inflammation; therefore, PBM at 35.7 J/cm^2^ can be used as the best energy dose associated with PDT [66].

Moreover, Santana-Blank et al. conducted a study to assess the serum levels of TNF-α, the soluble receptor (sIL-2R), and the distribution of peripheral leukocyte subgroups (CD4, CD8, and NK populations) in patients with advanced neoplastic disease undergoing infrared pulsed laser device (IPLD) treatment. A total of fifteen cancer patients (8 females and 7 males) were chosen with histologically confirmed and refractory carcinoma to conventional therapy. All the patients were subjected to IPLD treatment with a frequency between 0.5 MHz and 7.5 MHz. The selected subjects were further divided into two groups according to the outcome at the end of the clinical evaluation period: Those in group I were still alive and the patients in group II had died during the application of the protocol [67].

The IPLD treatment led to an increase in the initial TNF-α level in both groups; a decrease in TNF-α levels during the follow-up of group-I patients; a significant increase in serum sIL-2R levels in group-II patients compared to group-I patients; a progressive and steady increase in TNF-α levels in group-II; a decrease in the subpopulation of CD4+CD45RA+ in both groups and an increase in CD25+ cells; an increase in CD4+, CD4+CD45RA+, and CD25+ cells during the follow-up of group-II patients [67].

These results are consistent with the hypothesis that an inactive and necrotic regressing tumor is not able to produce TNF-α or to stimulate TNF-α production by other cells. Therefore, PBM can probably play an important role in restoring the regulatory mechanisms of the immune response that have been altered by tumor growth. Serum TNF-α levels may have potential value in the follow-up of tumor patients during PBM treatment and a decrease in TNF-α may be indicative of a better clinical response.

In addition, the results suggest that high levels of sIL-2R at the start of treatment may have a prognostic value in determining patients’ response to PBM. It is possible that treatment with PBM may produce changes in the host-to-tumor ratio due to a decrease in tumor load in patients who have reacted positively to it. On the contrary, the exaggerated production of this receptor could reflect an imbalance in the homeostasis of the T-cell subset and, in this way, PBM treatment is not able to control it. The authors concluded that IPLD treatment produces variations in the host-tumor relationship, due to a decrease in tumor load in those patients who reacted positively to it [67].

Some narrative reviews and case series that were not included in this review reported interesting results. Zecha et al. have shown that it seems unlikely that PBM has carcinogenic effects on normal cells or that it protects against the cytotoxic effects of radiotherapy (RT) since the non-ionizing wavelengths of the red and NIR spectrum used in PBM are much longer than the safety limit of 320 nm for DNA damage. There is also some evidence that suggests that PBM can improve treatment response. Further research is needed on the molecular pathways involved in PBM [14,15]. Recent investigations also highlight how PBM could increase treatment outcomes and progression-free survival in cancer patients [67,68,69,70,71,72].

The data seem to be particularly promising, since the side effects of many anticancer treatments often cause suffering and deeply influence these variables; in particular, quality of life has become increasingly important in technological evaluations and guidelines for the treatment of neoplasms. In addition, it was observed that there was a significant increase in cytomorphological changes associated with programmed cell death in neoplastic cells and, at the same time, no apparent changes were observed in non-neoplastic cells. These implications, with the others that have been mentioned, indicate a possible selective effect of PBM irradiation and how this may progressively affect the viability of tumor cells without initially interfering with their reproduction.

Even though many reports support the hypothesis that PBM may play an active role in the treatment of different neoplasms including the OSCCs, many questions remain unresolved. Firstly, while the biphasic dose–response is well established in normal tissues, it is not yet clear its role in malignancies. Secondly, while it seems that higher light fluencies create a cytotoxic amount of ROS that can directly destroy the tumor, it is crucial to clearly identify the right dosage that produce this positive effect. In other cases, the main effect of PBM seems to be the stimulation of the immune system, with low dosages that seems to be more effective. Subsequently, if the aim is to stimulate the immune system, it would be preferable to not to directly irradiate the tumor.

Finally, there were some limitations and considerations that should be acknowledged. First, the comparison process of variables and meta-analysis could not be performed, due to the lack of standardization of the methodological approaches, the diversity of PBM protocols, and the different sample types among the included studies. Second, only the articles in English language were considered in this review, which is considered a kind of selection bias.

## 5. Conclusions

With taking into consideration the limitations of this study, it is possible to hypothesize that there are possibilities for PBM to play a beneficial role in treating cancer patients. This review may stimulate the researchers and investigators to pursue further studies on oral cancer to study the biological action and clinical efficacy of PBM, and to identify the safe and correct protocols and dosages.

## Figures and Tables

**Figure 1 healthcare-09-00134-f001:**
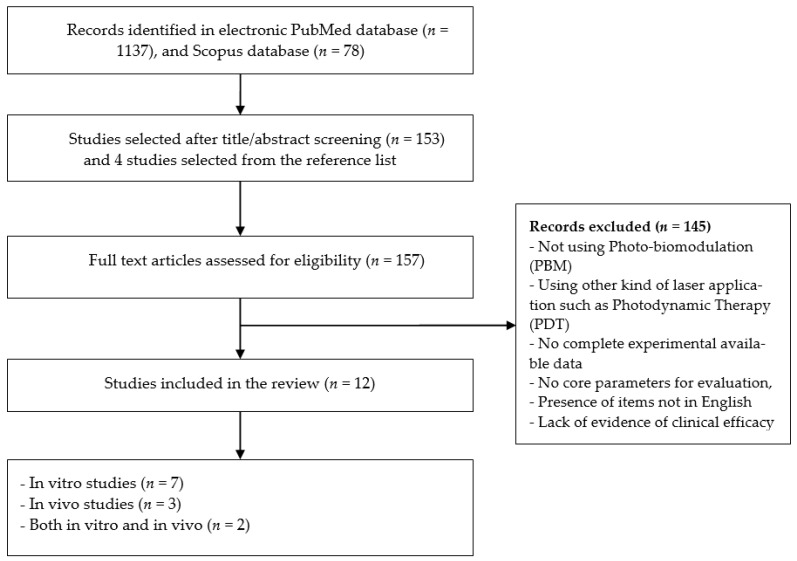
Flow diagram of the evaluation process of publications.

**Figure 2 healthcare-09-00134-f002:**
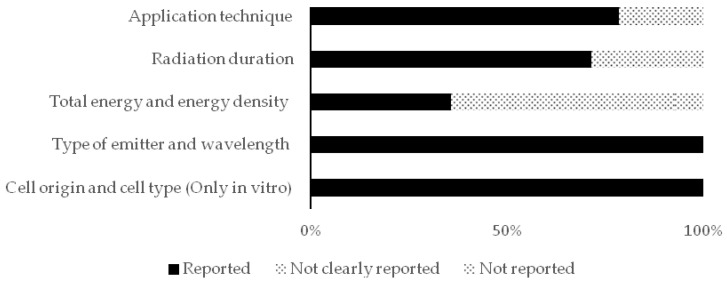
Reporting quality scores in all the included studies (*n* = 12) presented in percentages of articles (% of total).

**Figure 3 healthcare-09-00134-f003:**
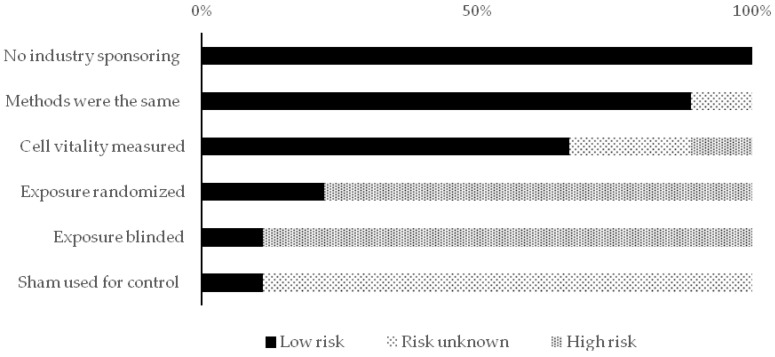
Risk of bias assessment scores in the in vitro studies (*n* = 9) presented in percentage of articles (% of total).

**Figure 4 healthcare-09-00134-f004:**
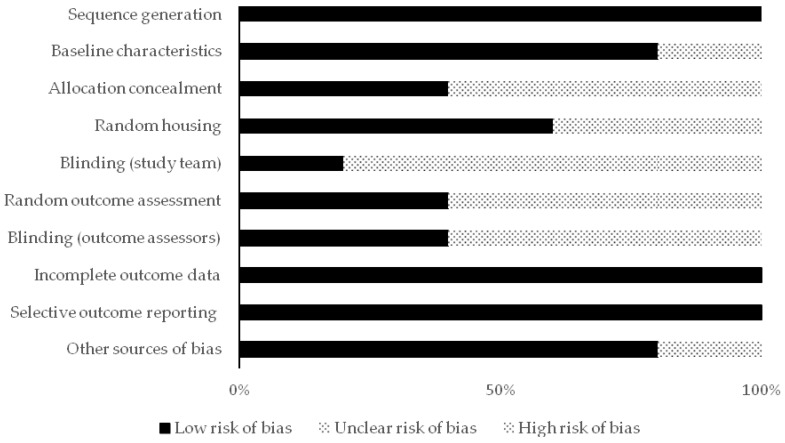
Risk of bias assessment scores in the in vivo studies (*n* = 5) with SYRCLE’s tool presented in percentage of articles (% of total).

**Table 1 healthcare-09-00134-t001:** Reporting quality scheme (for both in vitro and in vivo studies) and the risk of bias scheme (Only for in vitro studies) [21].

Assessment Type	Item
**Reporting quality** (both in vivo and in vitro studies)	- Is the cell origin and cell type used reported? *
	- Are the type of emitter and wavelength reported?
	- Are the total energy and energy density reported?
	- Is the radiation duration of PBM reported?
	- Is the application technique adequately reported?
**Risk of bias scheme** (only in vitro studies)	
- Performance bias	- Is a sham used for control treatment?
	- Was the exposure blinded?
	- Was the exposure randomized?
- Selection bias	- Is the cell vitality scored/measured?
- Detection bias	- Were the methods the same for control and exposure treatment?
- Other bias	- Was there no industry sponsoring involved?

* This item was only used for in vitro studies.

**Table 2 healthcare-09-00134-t002:** A summarized description of all the included studies in this review.

Author, Year	Study Type	Sample	Laser	Method of Evaluation	Main Outcomes	Conclusions	PBM Effect
1. Sroka et al., 1999 [11]	In vitro	Human SCC ^1^ of gingival mucosa (ZMK1) and other human cells	Different lasers; 410, 488, 630, 635, 640, 805, and 1064 nm	Mitosis rate by orcein-staining and cell proliferation by BrdU-test	A slight decrease in the mitotic rate of ZMK1 was observed with the increase in the irradiation energy independently with the wavelength. At irradiation of 20 J/cm^2^, a slight decrease in mitosis rate was observed when compared to controls without dependence on wavelength.	At specific parameters, an inhibitory effect of PBM was observed on human SCC when compared to controls.	Inhibitory
2. de Castro et al., 2005 [22]	In vitro	KB cells	Diode lasers 685 and 830 nm	Cellular viability by MTT spectroscopy assay	The time significantly influenced the cellular viability in both control and test groups (for both wavelengths). The PBM in both test groups significantly influenced the cellular viability when compared to control. The test group irradiated with 830 nm showed a significant increase in proliferation when compared to the other test group (685 nm).	PBM had a significant bio-stimulatory effect on KB cells proliferation influenced by the wavelength.	Stimulatory
3. Frigo et al., 2009 [23]	In vitro/In vivo	Melanoma cells (B16F10)/Melanoma cells in mouse model	InGaAlAsP ^2^ laser 660 nm	In vitro: cell viability and cell cycle changes by Tripan Blue, MTT, and cell quest histogram. In vivo: tumor volume and histological characteristics.	In vitro: The high irradiance (2.5 W/cm^2^) combined with high dose (1050 J/cm^2^) stimulated melanoma tumor growth. In vivo: A significant increase in the tumor volume, blood vessels and cell abnormalities was observed in the group of does 1050 J/cm^2^.	PBM over melanoma showed a stimulative effect and increase in tumor growth when applied in high irradiance and dose.	Stimulatory
4. de C Monteiro et al., 2011 [24]	In vivo	Cancerous lesions on hamster’s cheek induced by chemical carcinogenesis	Diode laser 660 nm	Histological analysis	The test group (with PBM) showed a significant difference in the amount of poorly differentiated tumors when compared to other groups without PBM.	PBM with these parameters may cause a progression of the severity of oral SCC in hamsters.	Stimulatory
5. Myakishev-Rempel et al., 2012 [25]	In vivo	SKH mouse nonmelanoma UV ^3^-induced skin cancer model	NASA LED ^4^ 670 nm	Photographic measurements of tumor growth	PBM didn’t show a measurable effect on tumor growth.	PBM with these parameters may be safe in case of application in presence of malignant lesions.	Inhibitory
6. Schartinger et al., 2012 [26]	In vitro	Human SCC cells (SCC25) and human normal cells	GaAlAs ^5^ 660 nm	Cell proliferation assay by MTT, cell cycle analysis, and apoptosis assay	In SCC25 cells, PBM showed a significant decrease in cell proliferation and in the percentage of G1-phase cells, and a significant increase in the percentage of S-phase cells when compared to the control. PBM showed a proapoptotic effect in SCC25.	PBM with these parameters did not show a stimulative effect.	Inhibitory
7. Sperandio et al., 2013 [12]	In vitro	Oral dysplastic cells (DOK) and oral cancer cells (SCC9 and SCC25)	GaAlAs laser 660 nm and 780 nm	Cellular viability by 3-h MTS assay, the apoptosis rate by TUNEL assay, and proteins analysis by Western blot and immunofluorescence	In SCC9, PBM showed inhibition of growth with 660 nm and stimulative effect with 780 nm. In SCC25, PBM showed a stimulative effect with both wavelengths. At 72 h evaluation time, PBM showed the lower levels of stimulation. PBM showed an effect on proteins and in particular caused an increased expression of p-Akt ^6^, pS6 and cyclin D1 proteins producing an aggressive isoform of Hsp90. Only SCC25 showed apoptosis when irradiated with 780 nm at 48 h (6.15 J/cm^2^) and at 72 h (3.07 J/cm^2^).	PBM with these parameters can aggravate oral cancer cellular behavior and modify the expression of proteins related to the progression and invasion of cancer cells.	Stimulatory
8. Gomes Henriques et al., 2014 [27]	In vitro	Human SCC of tongue (SCC25)	InGaAlP ^7^ laser 660	Cell growth assay, cell invasion analysis by Matrigel assay, and protein expression analysis	PBM on SCC25 with energy density of 1.0 J/cm^2^ showed a significant increase in proliferation, and expression of cyclin D1 and nuclear β-catenin, and a promotion of invasion through the reduction of E-cadherin and induction of MMP-9 ^8^ expression.	PBM stimulated the proliferation and invasion of SCC25 and caused alterations on proteins expression.	Stimulatory
9. Ottaviani et al., 2016 [18]	In vitro/In vivo	In vitro: Mouse melanoma cells (B16F10) and other human cells In vivo: Oral carcinogenesis model with 4-NQO ^9^ on mouse tongue	GaAs ^10^ and InGaAlAsP lasers 660, 800, and 970 nm	In vitro: ATP production assay In vivo: Histological evaluation and Immunofluorescence, real time PCR ^11^ and Flow Cytometry	In vitro: PBM showed an increase in cellular metabolism. In vivo: PBM reduced the tumor progression and this was associated with secretion of type I interferons from T lymphocytes and dendritic cells. A decrease in the angiogenic macrophages was observed in the tumor mass with a promotion of the vessel’s normalization.	PBM reduced tumor growth and was a safe procedure.	Inhibitory
10. Rhee et al., 2016 [28]	In vivo	Anaplastic thyroid cancer cell line FRO in mouse model	Diode laser 650 nm	Tumor volume, histological evaluation, and IHC ^12^ staining analysis	PBM caused an elevation of HIF-1α ^13^ and p-Akt, and a decrease in TGF-β1 ^14^ expression that play a role in the cell cycle regulation.	These effects may cause an over-proliferation and angiogenesis of cancer cells. PBM may cause aggressiveness of cancer through TGF-β1 and Akt/HIF-1α cascades.	Stimulatory
11. Takemoto et al., 2019 [29]	In vitro	Human SCC cell line (CAL27) seeded over normal stromal gingival fibroblasts	LED 660 nm	Expansion of colonies and cell counts, viability and apoptosis after PBM with 36 J/cm^2^	After 72 h of treatment, PBM inhibited the expansion of colonies. At high dose (36 J/cm^2^), PBM showed a general advantage with regard to the cell viability, apoptosis, and death assays on the stromal fibroblasts over cancer cells.	PBM (LED) at high doses inhibited in vitro the progression and number of cancer cells colonies without affecting the surrounding fibroblasts.	Inhibitory
12. Shirazian et al., 2020 [30]	In vitro	SCC cells originated from tongue (TSCC-1)	Diode lasers 660 and 810 nm	Cell proliferation by MTT assay, and flow cytometry to assess cyclin D1, β-catenin, E-cadherin, and MMP-9 markers. RT-PCR ^15^ to assess Ki67 and VEGF ^16^ expression levels	At 24 h evaluation, the cell proliferation was generally lower in PBM groups. In 810 nm groups (100 and 200 mW), higher percentages of cyclin D1and MMP-9 were observed, and a significant decrease in VEGF marker in the 810 nm group of 200 mW. In 660 nm groups (40 and 80 mW), higher percentages of β-catenin and E-cadherin were observed. No differences were observed among groups for the Ki67 marker.	PBM with 660 nm (80 mW) and 810 nm (200 mW) showed a significant inhibitory effect on cell proliferation at 0 and 24 h.	Inhibitory

^1^ Squamous Cell Carcinoma (SCC); ^2^ Indium–Gallium–Alluminium-Arsenide Phosphide (InGaAlAsP); ^3^ Ultraviolet (UV); ^4^ Light-Emitting Diode (LED); ^5^ Gallium Aluminium Arsenide (GaAlAs); ^6^ phospho Akt (p-Akt); ^7^ Indium–Gallium–Aluminum Phosphide (InGaAlP); ^8^ Matrix Metallopeptidase-9 (MMP-9); ^9^ 4-nitroquinoline-1-oxide (4-NQO); ^10^ Gallium Arsenide (GaAs); ^11^ Polymerase Chain Reaction (PCR); ^12^ Immunohistochemical (IHC); ^13^ Hypoxia Inducible Factor-1α (HIF-1α); ^14^ Transforming Growth Factor-β1 (TGF-β1); ^15^ Real-Time Polymerase Chain Reaction (RT-PCR); ^16^ Vascular Endothelial Growth Factor (VEGF).

**Table 3 healthcare-09-00134-t003:** Parameters of Photo-biomodulation (PBM) of all the included studies in this review.

Author; Tear	Wavelength	Type of Emitter	Power (mW)	Energy per Point (J)	Irradiation Time (sec)	Spot Size (cm^2^)	Energy Density (J/cm^2^)	Total Energy (J)	PBM Technique	PBM Schedule
1. Sroka et al., 1999 [11]	410, 488, 630, 635, 640, 805, and 1064 nm	Kr^+^, Ar^+^- pumped tunable dye, GaAlAs ^1^, and Nd:YAG ^2^ lasers	–	–	–	–	0–20	–	–	–
2. de Castro et al., 2005 [22]	685 nm 830 nm	Diode lasers	31 34.5	–	–	0.8	4	–	–	2 sessions with 48 intervals
3. Frigo et al., 2009 [23]	660 nm	InGaAlAsP ^3^	50	3 21	60 420	0.02	150 1050	9 63	CW ^4^	3 sessions with 24 h intervals
4. de C Monteiro et al., 2011 [24]	660 nm	Diode laser	30	4	133	0.07	56.4	–	CW	Every other day for 4 Weeks
5. Myakishev-Rempel et al., 2012 [25]	670 nm	NASA LED ^5^	–	–	312	–	2.5	–	–	2 sessions daily for 37 days
6. Schartinger et al., 2012 [26]	660 nm	GaAlAs	350	–	900	–	–	–	–	3 sessions with 24 h intervals
7. Sperandio et al., 2013 [12]	660 nm 780 nm	GaAlAs	40	–	–	0.039	2.05 3.07 6.15 (for each)	–	In contact	One session
8. Gomes Henriques et al., 2014 [27]	660 nm	InGaAlP ^6^	30	0.48 0.99	16 33	0.03	0.5 1.0	–	CW	2 sessions with 48 h intervals
9. Ottaviani et al. 2016 [18]	660 nm 800 nm 970 nm	GaAs ^7^ InGaAlAsP	100 1 W 2.5 W	–	60 30 30	–	3 6 6	–	CW	In vivo: one session a day for 4 days
10. Rhee et al., 2016 [28]	650 nm	Diode laser	–	0.3 0.6	150 300	0.02	15 30	0.3 0.6	In contact and CW	One session
11. Takemoto et al., 2019 [29]	660 nm	LED	100	–	–	–	3 6 9 12 24 36	–	–	3 times with 24 h intervals
12. Shirazian et al., 2020 [30]	660 nm 810 nm	Diode laser	40, 80 100, 200	–	30, 15 12, 6	0.3	4	–	Non-contact and CW	4 session with 0, 24, 72, and 168 h intervals

^1^ Gallium Aluminium Arsenide (GaAlAs); ^2^ Neodymium-doped Yttrium Aluminum Garnet (Nd:YAG); ^3^ Indium–Gallium–Alluminium-Arsenide Phosphide (InGaAlAsP); ^4^ Continuos Wave (CW); ^5^ Light-Emitting Diode (LED); ^6^ Indium–Gallium–Aluminum Phosphide (InGaAlP); ^7^ Gallium Arsenide (GaAs).

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
