# Peer review of "Laser Photobiomodulation (PBM)—A Possible New Frontier for the Treatment of Oral Cancer: A Review of In Vitro and In Vivo Studies"

_healthcare, 2021, doi:10.3390/healthcare9020134_

Round 1
Reviewer 1 Report
The reviewer understands the authors’ opinion. However, the review is not systemic review and the PICO is inappropriate.
If the review process is continued, the reviewer suggests;
1) Please change “systematic review” to “review”.
2) Please delete the comments about PICO model and related parts.
3) Please change the title to “Laser Photobiomodulation (PBM)—a possible new frontier for the treatment of oral cancer: A review of in vitro and in vivo studies.”
Author Response
Dear Editor,
We would like to thank you and your reviewers for taking time to review our manuscript and for the valuable comments. I am pleased to resubmit our major revised manuscript entitled “Laser Photobiomodulation (PBM)—a possible new frontier for the treatment of oral cancer: A review of in vitro and in vivo studies.” for consideration by healthcare Journal with a manuscript ID of healthcare-1083307. The responses to the reviewer’s comments are presented below point by point. All changes in the manuscript were highlighted by the "Track Change" function as recommended.
Kind regards,
Gaspare Palaia
Response to reviewer comment:
The reviewer understands the authors’ opinion. However, the review is not systemic review and the PICO is inappropriate.
If the review process is continued, the reviewer suggests;
Comment #1: 1) Please change “systematic review” to “review”.
Response: The “systematic review” was changed into “ review” as recommended in the title and manuscript.
Comment #2: 2) Please delete the comments about PICO model and related parts.
Response: All the paragraphs related to the PICO model were deleted.
Comment#3: 3) Please change the title to “Laser Photobiomodulation (PBM)—a possible new frontier for the treatment of oral cancer: A review of in vitro and in vivo studies.”
Response: The title was changed into “Laser Photobiomodulation (PBM)—a possible new frontier for the treatment of oral cancer: A review of in vitro and in vivo studies.”.
Reviewer 2 Report
The study is interesting, however it presents many methodological problems.
The studies considered are inhomogeneous, in fact the authors report "There were a diversity among the selected studies of the type of used laser, the PBM parameters, and the methodological approach, and the sample type and size".
Because of the extreme inhomogeneous of the studies taken into consideration they were divided , from the authors, in subgroups still inhomogeneous and then not comparable each other.
The methods used often appear confusing and follow the aim of the study is difficult. Furthermore the results are reported in a unclear manner.
The studies considered are reported , in the results section, in unclear and imprecise way, this make difficult to understand them.
Finally, from the discussion it seems that the review does’nt give an answer to the authors’ PICO question
Author Response
Dear Editor,
We would like to thank you and your reviewers for taking time to review our manuscript and for the valuable comments. I am pleased to resubmit our major revised manuscript entitled “Laser Photobiomodulation (PBM)—a new frontier for the treatment of oral cancer. Biological principles and future perspectives. Systematic review.” for consideration by healthcare Journal with a manuscript ID of healthcare-1083307. The responses to the reviewer’s comments are presented below point by point. All changes in the manuscript were highlighted by the "Track Change" function as recommended.
Kind regards,
Gaspare Palaia
Response to reviewer comments:
Comment#1: The study is interesting, however it presents many methodological problems.
The studies considered are inhomogeneous, in fact the authors report "There were a diversity among the selected studies of the type of used laser, the PBM parameters, and the methodological approach, and the sample type and size".
Because of the extreme inhomogeneous of the studies taken into consideration they were divided , from the authors, in subgroups still inhomogeneous and then not comparable each other.
The methods used often appear confusing and follow the aim of the study is difficult. Furthermore the results are reported in a unclear manner.
The studies considered are reported , in the results section, in unclear and imprecise way, this make difficult to understand them.
Response: A comprehensive revision has been made to the methods, results, and discussion sections. The aim of the study was changed in the way to be restricted to studies that investigate the PBM effect on cancer, and correspondingly some modifications were done to follow the aim in the methods section. While for the results section, the included studies in the review have been reduced according to the new aim. Also, two new tables have been constructed, with the elimination of the old tables, and a comprehensive revision has been made to the way of reporting the results.
Comment#1: Finally, from the discussion it seems that the review does’nt give an answer to the authors’ PICO question
Response: The “PICO” model seemed to be inappropriate for this study therefore it has been removed with its related parts of the manuscript as recommended. This experience revealed that there is no-till now complete approval about the effect of PBM on the neoplastic site. We hope with this study to encourage performing more studies on this topic to achieve more concrete results.
Reviewer 3 Report
The paper is an interesting review of the use of laser photobiomodulation on oral cancer. It is very interesting, I enjoyed reading it. In my opinion, this article is eligible for publication, as it is very well written and complete.
The main limitation of this paper is the fact that the various paper analyzed are heterogeneous, so no metanalysis to better evaluate the use of photobiomodulation could be made.
Also, the topic is very ambiguous, in the sense that a lot of lights have been tried, and the results obtained were very variable. I would probably add a part to discuss the absorption spectra of the various lights.
In the introduction you could also state that "photobiomodulation is already in use in the treatment of various medical condition, especially in dermatology" and cite an article such as :" Cannarozzo G, Silvestri M, Tamburi F et al. A new 675-nm laser device in the treatment of acne scars: an observational study. Lasers Med Sci. 2021 Feb;36(1):227-231."
The article is highlighted in some parts, so I guess this version has already been reviewed. I would however suggest the authors to remove all highlighted parts.
Thank you
Author Response
Dear Editor,
We would like to thank you and your reviewers for taking time to review our manuscript and for the valuable comments. I am pleased to resubmit our major revised manuscript entitled “Laser Photobiomodulation (PBM)—a new frontier for the treatment of oral cancer. Biological principles and future perspectives. Systematic review.” for consideration by healthcare Journal with a manuscript ID of healthcare-1083307. The responses to the reviewer’s comments are presented below point by point. All changes in the manuscript were highlighted by the "Track Change" function as recommended.
Kind regards,
Gaspare Palaia
Response to reviewer comment:
Comment#1: The paper is an interesting review of the use of laser photobiomodulation on oral cancer. It is very interesting, I enjoyed reading it. In my opinion, this article is eligible for publication, as it is very well written and complete.
The main limitation of this paper is the fact that the various paper analyzed are heterogeneous, so no metanalysis to better evaluate the use of photobiomodulation could be made.
Also, the topic is very ambiguous, in the sense that a lot of lights have been tried, and the results obtained were very variable. I would probably add a part to discuss the absorption spectra of the various lights.
Response: Thank you for the valuable comments. A comprehensive revision was made in the results and discussion sections, and a paragraph has been added to discuss the absorption spectra of various lights in the discussion section (Page 3).
Comment#2: In the introduction you could also state that "photobiomodulation is already in use in the treatment of various medical condition, especially in dermatology" and cite an article such as :" Cannarozzo G, Silvestri M, Tamburi F et al. A new 675-nm laser device in the treatment of acne scars: an observational study. Lasers Med Sci. 2021 Feb;36(1):227-231."
Response: A modification has been made in the introduction section adding this recommended paragraph with reference. The paragraph becomes “ Even though studies on PBM began in the early 1960s and a large number of them describe its beneficial effects in the treatment of various medical conditions, especially in dermatology [10], many aspects of the clinical use of PBM are still only partially understood, and, to date, the same limits apply in terms of its therapeutic possibilities.”
Comment#3: The article is highlighted in some parts, so I guess this version has already been reviewed. I would however suggest the authors to remove all highlighted parts.
Response: The highlights have been removed and all the modifications have been highlighted with the “Track change” function.
Round 2
Reviewer 1 Report
The revision was acceptable.
Reviewer 2 Report
The manuscript, after revision, has been significantly improved and is currently adequate for publication.
This manuscript is a resubmission of an earlier submission. The following is a list of the peer review reports and author responses from that submission.
Round 1
Reviewer 1 Report
The review shows that there may be the effects of PBM on OSCC.
This is an interesting study. However, I would like to make some points regarding the manuscript. The review needs to be revised. First, please revise the text following the PRISMA guideline. Second, the PICO is inappropriate.
Comments in detail:
INTRODUCTION
1) Please delete the duplicate (1.1 Rationale and 1.2 Objectives) (L92-115).
2) Please revise PICO and add clinical question. If “O” is positive effects of PBM, “I” should be PBM and “C” be no PBM.
3) Please provide an explicit statement of the question(s) the review will address with reference to participants, interventions, comparators, and outcomes (PICO).
MATERIALS AND METHODS
1) Please revise the text following the PRISMA guideline.
2) Please indicate if a review protocol exists, if and where it can be accessed (e.g., Web address), and, if available, provide registration information including registration number.
3) Please describe the month of 2019 (L140) and date last searched in addition to L145.
4) The authors only mentioned “AND” for keywords. The authors should use “OR”, too.
5) Please describe method of data extraction from reports (e.g., piloted forms, independently, in duplicate) and any processes for obtaining and confirming data from investigators. Please also add the name of investigators.
6) If the authors make a systematic review, they should not include in vitro, in vivo study. Furthermore, based on new PICO, please re-consider whether the review include systematic, narrative, case series or not.
7) Please describe methods used for assessing risk of bias of individual studies (including specification of whether this was done at the study or outcome level), and how this information is to be used in any data synthesis.
8) Please state the principal summary measures (e.g., risk ratio, difference in means).
9) Please describe the methods of handling data and combining results of studies, if done, including measures of consistency (e.g., I2) for each meta-analysis.
10) Please specify any assessment of risk of bias that may affect the cumulative evidence (e.g., publication bias, selective reporting within studies).
11) Please describe methods of additional analyses (e.g., sensitivity or subgroup analyses, meta-regression), if done, indicating which were pre-specified.
RESULTS
1) Please check the plagiarism using a software (L176-628) and delete inappropriate parts and revise this part following the PRISMA guideline. Some references should be moved to the discussion part.
2) New results will be added and completely revise this section following the PRISMA checklist.
DISCUSSION SECTION
1) Please change the discussion following the new results.
2) Please discuss limitations at study and outcome level (e.g., risk of bias), and at review-level (e.g., incomplete retrieval of identified research, reporting bias).
3) Please revise the conclusion following a PICO.
Author Response
Dear Editor,
We would like to thank you and your reviewers for taking time to review our manuscript and for the valuable comments. I am pleased to submit our major revised manuscript entitled “Laser Photobiomodulation (PBM)—a new frontier for the treatment of oral cancer. Biological principles and future perspectives. Systematic review.” for consideration by healthcare Journal with a manuscript ID of healthcare-1006410. The responses to the reviewer’s comments are presented below point by point. All changes in the manuscript were highlighted by the "Track Change" function as recommended.
Kind regards,
Gaspare Palaia
Response to reviewer comment:
Comment #1: The review shows that there may be the effects of PBM on OSCC.
This is an interesting study. However, I would like to make some points regarding the manuscript. The review needs to be revised. First, please revise the text following the PRISMA guideline. Second, the PICO is inappropriate.
Response: Thank you for the valuable and constructive comments. A comprehensive revision has been made for the entire text of the study.
Comments in detail:
- INTRODUCTION
Comment #2: 1) Please delete the duplicate (1.1 Rationale and 1.2 Objectives) (L92-115).
Response: The titles of these sections and the duplicated sentences were eliminated (Page 2, Line 80-93).
Comment #3: 2) Please revise PICO and add clinical question. If “O” is positive effects of PBM, “I” should be PBM and “C” be no PBM.
Response: A revision has been made for this section and the PICO question has been added “Through observing the reported PBM-induced cellular responses (C), can PBM application (I) be introduced as a safe and beneficial adjunctive (O) to the conventional treatments in patients with oral cancer (P)?” (Page 3, Line 95-98).
Comment #4: 3) Please provide an explicit statement of the question(s) the review will address with reference to participants, interventions, comparators, and outcomes (PICO).
Response: This PICO question has been addressed with referring to each component of PICO (Page 3, Line 95-98).
- MATERIALS AND METHODS:
Comment #5: 1) Please revise the text following the PRISMA guideline.
Response: A major revision has been made to all the methods section.
Comment #6: 2) Please indicate if a review protocol exists, if and where it can be accessed (e.g., Web address), and, if available, provide registration information including registration number.
Response: The protocol of this systematic review has been submitted for registration at PROSPERO with ID number 224772. This point has been added and described in the text (page 3, Line 112-114).
Comment #7: 3) Please describe the month of 2019 (L140) and date last searched in addition to L145.
Response: This research has been conducted starting in September 2018 and the last search was performed in March 2020. This details has been added to the manuscript on page 3, line 119-120.
Comment #8: 4) The authors only mentioned “AND” for keywords. The authors should use “OR”, too.
Response: This was an error, where “OR” was used in the search. It has been added to the text (Page 3, Line 118)
Comment #9: 5) Please describe method of data extraction from reports (e.g., piloted forms, independently, in duplicate) and any processes for obtaining and confirming data from investigators. Please also add the name of investigators.
Response: The data extraction was carried out independently between two investigators and in case of disagreements, a discussion with the third investigator was performed. This description has been added to the text with the names of investigators (Page 3, Line 129-130).
Comment #10: 6) If the authors make a systematic review, they should not include in vitro, in vivo study. Furthermore, based on new PICO, please re-consider whether the review include systematic, narrative, case series or not.
Response: A revision has been made for this issue. Based on new PICO, systematic, narrative and case series were eliminated from the study (Page 3, Line 99-109).
Comment #11: 7) Please describe methods used for assessing risk of bias of individual studies (including specification of whether this was done at the study or outcome level), and how this information is to be used in any data synthesis.
Response: The methods used for assessing the reporting quality and risk of bias have been added and described in the revised version of the manuscript on page 4, Line 142-157.
Comment #12: 8) Please state the principal summary measures (e.g., risk ratio, difference in means).
Response: The principal summary measures was no performed. Only assessment of individual studies and a narrative review based on this systematic approach were performed.
Comment #13: 9) Please describe the methods of handling data and combining results of studies, if done, including measures of consistency (e.g., I2) for each meta-analysis.
Response: A narrative and tabular synthesis of data were performed for all the studies including; the author/year, the study design, the characteristics of the samples, laser type and protocol, and PBM application outcome. The outcomes of the included studies were divided into two categories as following; 1) studies that demonstrated the general effect of PBM on tumor cells, and 2) studies that described possible biological mechanisms of PBM on tumor cell. The later was further divided into three subcategories including; 1) studies that demonstrated the direct effect of light on tumor cells, 2) studies that demonstrated the possible differential effect of PBM on malignant cancer cells and on normal healthy cells when combined with additional cytotoxic and antitumor therapies, and 3) studies that demonstrated the possible role of PBM in stimulating the immune system. This description has been added to the manuscript on page 3, Line 130-140.
The meta-analysis were not performed and this issue have been described in the methods (page 4, Line 158-159), results (Page 6, Line 185-188) and limitations of the study (Page 17, Line 493-498).
Comment #14: 10) Please specify any assessment of risk of bias that may affect the cumulative evidence (e.g., publication bias, selective reporting within studies).
Response: Only qualitative analysis (assessment of risk of bias in individual studies) was performed that included the selective reporting outcome in the selected in vivo studies by the SYRCLE’s tool. However, meta-analysis was not performed and a limitation of this review has been added to the manuscript for this issue. (Page 17, Line 493-498)
Comment #15: 11) Please describe methods of additional analyses (e.g., sensitivity or subgroup analyses, meta-regression), if done, indicating which were pre-specified.
Response: The analysis were not performed and this issue have been described in the methods (page 4, Line 158-159), results (Page 6, Line 185-188) and limitations of the study (Page 17, Line 493-498).
- RESULTS:
Comment #16: 1) Please check the plagiarism using a software (L176-628) and delete inappropriate parts and revise this part following the PRISMA guideline. Some references should be moved to the discussion part.
Response: A revision has been made for this part of the manuscript and a check of plagiarism using a software to confirm this issue.
Comment #17: 2) New results will be added and completely revise this section following the PRISMA checklist.
Response: A revision has been made and the included studies have been organised where the tables were modified according to the new PICO.
- DISCUSSION SECTION:
Comment #18: 1) Please change the discussion following the new results.
Response: A modification has been performed to the discussion section (Page 17, Line 461-474) and a paragraph of limitation was added (Page 17, Line 493-498).
Comment #19: 2) Please discuss limitations at study and outcome level (e.g., risk of bias), and at review-level (e.g., incomplete retrieval of identified research, reporting bias).
Response: A paragraph of limitations has been added to the discussion section on page 17, Line 493-498. “There were some limitations and considerations that should be acknowledged. First, the comparison process of variables and meta-analysis could not be performed, due to the lack of standardization of the methodological approaches, the diversity of PBM protocols, and the different sample types among the included studies. Second, only one database was considered for carrying out this review and only the articles in English language were considered in this review which are considered a kind of selection bias.”
Comment #20: 3) Please revise the conclusion following a PICO.
Response: A revision has been made to the conclusion section and it became “With taking into consideration the limitations of this study, it is possible to hypothesize that there are effective possibilities for PBM to play a beneficial role in treating cancer patients. This review may stimulate the researchers and investigators to pursue further studies on oral cancer to study the biological action and clinical efficacy of PBM, and to identify the safe and correct protocols and dosages.” (Page 17, Line 500-504).
Reviewer 2 Report
Dear editor and authors,
Thank you for giving me the opportunity to peer review the present manuscript. Although the systematic review submitted for evaluation is promising, certain not very robust aspects should be improved before its consideration for publication. Therefore, I cannot recommend its acceptance under its current state. The following constructive comments are recommended to the authors to methodologically improve their paper in future:
Title
- The title should be corrected. This was a “systematic review”. Nowadays the concept “literature review” directly indicates a narrative review. According to influential guidelines (AMSTAR, PRISMA, MOOSE, etc), the paper should include in the title if a systematic review (and meta-analysis if performed) was conducted.
Introduction
- An up to date and more explicit paragraph showing oral cancer epidemiology (minimally, number of cases and deaths per year) should be written and updated using one or two correct references (e.g., GLOBOCAN last report).
- The objectives are very different from the wide scope of the title. Also very generic, clinical outcomes, for example, should be more precisely and detailed described.
Methods (Here I found the main methodological issues that make this review unable to be accepted for publication):
“The methods and criteria for inclusion were selected based on the PRISMA Declaration, which offers a protocol with respect to the reference items that have been included in this systematic review.”
- One of the main issues is the absence of an a priori design of a study protocol. Authors followed PRISMA guidelines, but did not registered their protocol (for example in PROSPERO International Prospective register of systematic reviews protocols), and therefore, did not meet the “protocol” PRISMA item). The publication or registration of a systematic review/meta-analysis protocol is strongly encouraged by consensus guidelines, and meta-epidemiological studies are currently showing that these systematic reviews are seriously biased. Here, the results could be seriously biased if based on ad hoc and non-systematic approaches.
- The design was also aberrant in the selection of databases. MEDLINE and PubMed are the same (MEDLINE is accessed through PubMed), and Google scholar is not recommended as the main database, because it does not allow an appropriate repeatability of searches by other researchers. So, only one pertinent database was analyzed. Cochrane Collaboration strongly suggests the use of Embase (as a complement to PubMed). Finally, clinical trials registers or CENTRAL database (also from Cochrane) should also be searched. In summary, there is probably a potential selection bias.
- “no restrictions were imposed on the language of the primary studies”
This is not true, please, carefully revise your eligibility criteria: “The exclusion criteria were as follows: No experimental data available, Items not in English”
Restrictions were imposed, based on English language, and as authors know, it is another well-known source of selection bias.
- The identification and selection process of studies, following PRISMA guidelines, should be performed based on two different stages: first-titles and abstract, second-full text evaluation. Authors made this process in only one stage, and this is not recommended because it also significantly increases the error rate when selecting articles. The number of authors involved in this process and data extraction was not reported (nor inter-agreement scoring systems, e.g., kappa index).
- The qualitative analysis does not follow the standard recommended by the Newcastle Otawa scale. Furthermore, this scale is recommended for observational studies, and the aim of this review is to investigate experimental studies.
- Heterogeneity was not assessed or discussed, nor their sources. And publication bias analysis was also neglected.
- Finally, a limitations paragraph was not presented, when high impact journal (and, of course PRISMA guidelines), consider it mandatory.
In summary, I really recommend to the authors a new careful refocusing of this article, applying more solid and qualitative methods, registering an a priori protocol, and making results and conclusions with a higher quality of evidence. Authors should adhere more carefully to PRISMA standards. AMSTAR guidelines can also be followed for self-evaluation (the current manuscript presents a "critically low quality" according to AMSTAR2 scoring system), an appropriateness use of tools for risk of bias analysis (NOS scale or alternatives), and I also recommend them the measurement of quality of the evidence (e.g., GRADE system).
Author Response
Dear Editor,
We would like to thank you and your reviewers for taking time to review our manuscript and for the valuable comments. I am pleased to submit our major revised manuscript entitled “Laser Photobiomodulation (PBM)—a new frontier for the treatment of oral cancer. Biological principles and future perspectives. Systematic review.” for consideration by healthcare Journal with a manuscript ID of healthcare-1006410. The responses to the reviewer’s comments are presented below point by point. All changes in the manuscript were highlighted by the "Track Change" function as recommended.
Kind regards,
Gaspare Palaia
Response to reviewer comment:
Dear editor and authors,
Comment #1: Thank you for giving me the opportunity to peer review the present manuscript. Although the systematic review submitted for evaluation is promising, certain not very robust aspects should be improved before its consideration for publication. Therefore, I cannot recommend its acceptance under its current state. The following constructive comments are recommended to the authors to methodologically improve their paper in future:
Response: Thank you for the valuable and constructive comments. The manuscript has been revised and modified.
- Title
Comment #2: - The title should be corrected. This was a “systematic review”. Nowadays the concept “literature review” directly indicates a narrative review. According to influential guidelines (AMSTAR, PRISMA, MOOSE, etc), the paper should include in the title if a systematic review (and meta-analysis if performed) was conducted.
Response: The “Literature review” in the title has been changed to “systematic review” (Page 1, Line 4).
- Introduction
Comment #3: - An up to date and more explicit paragraph showing oral cancer epidemiology (minimally, number of cases and deaths per year) should be written and updated using one or two correct references (e.g., GLOBOCAN last report).
Response: A paragraph has been added to the manuscript describing the updated data on oral cancer epidemiology with appropriate references including the number of cases and deaths per year (Page 1, Line 37-41).
Comment #4: - The objectives are very different from the wide scope of the title. Also very generic, clinical outcomes, for example, should be more precisely and detailed described.
Response: A modification has been made to the objectives and becomes “The aim of this study is to systematically review the literature to observe the available data on PBM-induced cellular responses in cancer cells in vitro and in vivo, and directly in patients with carcinomas.” (Page 2, line 89-91).
Methods (Here I found the main methodological issues that make this review unable to be accepted for publication):
“The methods and criteria for inclusion were selected based on the PRISMA Declaration, which offers a protocol with respect to the reference items that have been included in this systematic review.”
Comment #5: - One of the main issues is the absence of an a priori design of a study protocol. Authors followed PRISMA guidelines, but did not registered their protocol (for example in PROSPERO International Prospective register of systematic reviews protocols), and therefore, did not meet the “protocol” PRISMA item). The publication or registration of a systematic review/meta-analysis protocol is strongly encouraged by consensus guidelines, and meta-epidemiological studies are currently showing that these systematic reviews are seriously biased. Here, the results could be seriously biased if based on ad hoc and non-systematic approaches.
Response: A comprehensive revision has been performed to the methods section. The protocol of this systematic review has been submitted for registration at PROSPERO with ID number 224772, and this point has been added and described in the text (page 3, Line 112-114).
Comment #6: - The design was also aberrant in the selection of databases. MEDLINE and PubMed are the same (MEDLINE is accessed through PubMed), and Google scholar is not recommended as the main database, because it does not allow an appropriate repeatability of searches by other researchers. So, only one pertinent database was analyzed. Cochrane Collaboration strongly suggests the use of Embase (as a complement to PubMed). Finally, clinical trials registers or CENTRAL database (also from Cochrane) should also be searched. In summary, there is probably a potential selection bias.
Response: A revision has been made to all the resulted studies from the performed search and the google scholar has been eliminated with its articles. PubMed becomes the only database searched in the revised version of the manuscript (Page 3, Line 114). Also, the absence of other important databases and the only use of one database were considered as a limitation of this study and this issue has been described in the manuscript (Page 17, Line 496).
Comment #7: - “no restrictions were imposed on the language of the primary studies”
This is not true, please, carefully revise your eligibility criteria: “The exclusion criteria were as follows: No experimental data available, Items not in English”
Restrictions were imposed, based on English language, and as authors know, it is another well-known source of selection bias.
Response: All the studies in other languages were excluded from the study. This was an error and a correction has been made to this issue (Page 3, Line 104-105 and Line 108). This issue has been described in the limitations of the study for consideration during the interpretation of the results (Page 17, Line 496-498).
Comment #8: - The identification and selection process of studies, following PRISMA guidelines, should be performed based on two different stages: first-titles and abstract, second-full text evaluation. Authors made this process in only one stage, and this is not recommended because it also significantly increases the error rate when selecting articles. The number of authors involved in this process and data extraction was not reported (nor inter-agreement scoring systems, e.g., kappa index).
Response: The study selection process was performed on two different stages (screening stage of titles and abstract, and full-text read stage) and modifications have been performed in the text (Page 3, Line 122-127) and flowchart (Page 5, Line 169) to describe this point better. The number of investigators, their names and the data extraction process have been better described (Page 3, Line 129-140).
Comment #9: - The qualitative analysis does not follow the standard recommended by the Newcastle Otawa scale. Furthermore, this scale is recommended for observational studies, and the aim of this review is to investigate experimental studies.
Response: The qualitative analysis by Newcastle Otawa scale (NOS) has been eliminated from the study. Different tools were used to perform the qualitative analysis (assessment of risk of bias in studies). A complete description of the methods of the used tools has been added to the manuscript (Page 4, line 141-162).
Comment #10: - Heterogeneity was not assessed or discussed, nor their sources. And publication bias analysis was also neglected.
Response: Modifications have been made and the diversity in the selected studies has been discussed in the results section (Page 5, Line 185-188) and being a cause of not performing the meta-analysis was acknowledged as a limitation to be considered in the discussion section (Page 17, Line 493-499).
Comment #11: - Finally, a limitations paragraph was not presented, when high impact journal (and, of course PRISMA guidelines), consider it mandatory.
Response: A paragraph of limitations has been added to the text on Page 17, Line 493-499).
Comment #12: In summary, I really recommend to the authors a new careful refocusing of this article, applying more solid and qualitative methods, registering an a priori protocol, and making results and conclusions with a higher quality of evidence. Authors should adhere more carefully to PRISMA standards. AMSTAR guidelines can also be followed for self-evaluation (the current manuscript presents a "critically low quality" according to AMSTAR2 scoring system), an appropriateness use of tools for risk of bias analysis (NOS scale or alternatives), and I also recommend them the measurement of quality of the evidence (e.g., GRADE system).
Response: The manuscript has been totally revised in particular the methods, results and conclusion sections. In this systematic review, the selected studies were in vivo and in vitro. Therefore, it was decided to use different tools for the qualitative analysis. The SYRCLE’s tool was used for the in vivo studies. Due to the absence of recommended risk of bias tool for in vitro studies in the literature, it was decided to create it based on the methods of a systematic review for in vitro studies (Page 4, Line 141-162). However, the meta-analysis were not performed. A description of this issues has been added to the manuscript (Page 4, Line 158-159, Page 6, Line 185-188, Page 17, Line 493-496).
Reviewer 3 Report
Although the topic is interesting, the revision is non very conclusive. Authors divide the results in three main sections. I suggest to first select one of the results to review and to reset the research focusing on one of it. In the present form, the review is quite confusing and doesn't lead to useful results.
The paper doesn't bring significant new knowledge in the field of interest.
So, this paper is not suitable for publication in the current form in this journal.
Author Response
Dear Editor,
We would like to thank you and your reviewers for taking time to review our manuscript and for the valuable comments. I am pleased to submit our major revised manuscript entitled “Laser Photobiomodulation (PBM)—a new frontier for the treatment of oral cancer. Biological principles and future perspectives. Systematic review.” for consideration by healthcare Journal with a manuscript ID of healthcare-1006410. The responses to the reviewer’s comments are presented below point by point. All changes in the manuscript were highlighted by the "Track Change" function as recommended.
Kind regards,
Gaspare Palaia
Response to reviewer comment:
Comment #1: Although the topic is interesting, the revision is non very conclusive. Authors divide the results in three main sections. I suggest to first select one of the results to review and to reset the research focusing on one of it. In the present form, the review is quite confusing and doesn't lead to useful results.
The paper doesn't bring significant new knowledge in the field of interest.
So, this paper is not suitable for publication in the current form in this journal.
Response: A comprehensive revision has been made to the manuscript in order to make it more focused.
Round 2
Reviewer 1 Report
The authors misunderstand the systematic review or PICO model. It treats human clinical studies.
Reviewer 2 Report
As I mentioned previously, the work presents a high risk of potential bias and presents serious methodological limitations. The authors have made modifications, but the article continues having a poor methodological design with a “critically low quality” score, according to AMSTAR2 guidelines.
The protocol has been recorded a posteriori, this completely biases the research. As I mentioned, the only way to alleviate the problem is to carry out a new investigation. I should also add that the protocol has not yet been accepted by PROSPERO -no solution-
The single use of PubMed is another relevant error, not complying with the PRISMA and AMSTAR2 standards and not following the recommendations of the COCHRANE collaboration. The article is consequently biased, with poor methodology that does not meet high standards -not solved-.
The language restriction also biases the results. -not solved-.
Finally, many aspects were not taken into consideration, for example, the quality of the evidence was not measured using GRADE system.
Reviewer 3 Report
Authors answered to the previous comments, improving the manuscript